🔓 | **Open Peer Review** | Environmental Microbiology | Research Article

# Rice bacterial leaf blight drives rhizosphere microbial assembly and function adaptation

Hubiao Jiang,[1] Jinyan Luo,[2] Quanhong Liu,[1] Solabomi Olaitan Ogunyemi,[1] Temoor Ahmed,[1] Bing Li,[1] Shanhong Yu,[3] Xiao Wang,[4] Chenqi Yan,[5] Jianping Chen,[6] Bin Li[1]

**ABSTRACT**  Rice bacterial leaf blight (BLB) is the most destructive phyllosphere bacterial disease caused by *Xanthomonas oryzae* pv. *oryzae*. The effect of soil-borne diseases on the rhizosphere microbiome has been extensively investigated. However, the regulatory role of the phyllosphere disease BLB on the rhizosphere microbiome remains unclear. Here, we observed that BLB significantly altered the composition of the bacterial-fungal community in the rhizosphere, decreasing bacterial diversity but not fungal diversity. The scale of inter-kingdom networks in the rhizosphere microbiome of BLB-infected plants was more complex and broader than in healthy plants, while the bacterial community was more vulnerable to BLB than the fungal community. Indeed, the relative abundance of *Streptomyces*, *Chitinophaga*, *Sphingomonas*, and *Bacillus* was higher in the BLB rhizosphere, which can be explained by their keystone hub taxa status in the rhizosphere co-occurrence network. Null model analysis showed that the deterministic assembly of bacterial communities increased while that of fungi decreased. Additionally, the assembly of bacterial and fungal communities was significantly related to variations in BLB and soil nutrients, especially pH and available phosphorus. Random forest model results showed that the bacterial community in the rhizosphere had a strong potential for predicting BLB. Metagenomic analysis revealed that BLB had a considerable impact on the functional adaptation of the rhizosphere microbiome. Interestingly, the abundance of some functional genes involved in carbon, phosphorus, and methane metabolism increased drastically.

**IMPORTANCE**  Our results suggest that rhizosphere bacteria are more sensitive to bacterial leaf blight (BLB) than fungi. BLB infection decreased the diversity of the rhizosphere bacterial community but increased the complexity and size of the rhizosphere microbial community co-occurrence networks. In addition, the relative abundance of the genera *Streptomyces*, *Chitinophaga*, *Sphingomonas*, and *Bacillus* increased significantly. Finally, these findings contribute to the understanding of plant-microbiome interactions by providing critical insight into the ecological mechanisms by which rhizosphere microbes respond to phyllosphere diseases. In addition, it also lays the foundation and provides data to support the use of plant microbes to promote plant health in sustainable agriculture, providing critical insight into ecological mechanisms.

**KEYWORDS**  rice bacterial leaf blight, rhizosphere microbiome, microbiome assembly, metagenomics

The interaction between the plant microbiome and the host plant plays a critical role in plant function and fitness in nature (1–5), which also serves as an essential factor in influencing plant health (6–8). Rhizosphere microorganisms contribute significantly to plant growth, nutrient acquisition, abiotic, and biotic stress (disease suppression)

Address correspondence to Jianping Chen, jpchen2001@126.com, or Bin Li, libin0571@zju.edu.cn.

The authors declare no conflict of interest.

See the funding table on p. 16.

(9–15). The use of plant microbial communities to promote crop yields and manage plant diseases is a promising approach to sustainable agriculture (16–19). The plant rhizosphere ecosystem is affected by many variables, which can be categorized as biotic or abiotic, such as the host type (20, 21), soil type (22, 23), disease invasion, and insect damage (24).

In recent years, there has been increasing evidence that pathogen invasion affects microbial communities (25, 26). For example, the occurrence of different degrees of potato scab caused by *Streptomyces* spp. can significantly affect the microbial community composition and function of potato topsoil (geocaulosphere soil), demonstrating that specific inhibitory effects on pathogens were significantly enriched (25). Moreover, the rhizosphere and root interior of *F. graminearum*-infected wheat were significantly enriched for *Stenotrophomonas* (SR80), a beneficial bacterium that induces robust disease resistance by enhancing part of the plant's defense responses (26). Recent research findings have documented that increasing the level of jasmonate in *Arabidopsis* helps to recruit beneficial microbial communities to enhance plant disease resistance (27). The research carried out by Lee et al. (28) on tomato bacterial wilt discovered that the beneficial bacteria of rhizosphere microorganisms decrease bacterial wilt disease by stimulating the plant's immune response (28). *Flavobacterium* (TRM1), which prevents tomato bacterial wilt, was isolated from the rhizosphere of soil resistant to tomato bacterial wilt (29). Hence, according to the above findings, it can be concluded that rhizosphere microbes contribute significantly to resistance to disease.

Rice (*Oryza sativa* L.) is the world's most important food crop; however, *Xanthomonas oryzae* pv. oryzae, which causes bacterial leaf blight (BLB), one of the most important rice crop diseases, reduces its yield and consequently threatens food security (30). *Xanthomonas* is a Gram-negative bacterium and is one of the most important genera in plant bacteriology with 40 species of the bacteria able to infect approximately 400 plant species. It enters rice plants through wounds or water holes and settles rapidly on the xylem of leaves, hence, generating gray-brown lesions along the veins (31). The above-ground pathogen infection has been demonstrated to influence the plant rhizosphere microbial community (27, 32). However, the rhizosphere microbial community's response mechanism to BLB remains unclear. As a consequence, we collected rice rhizosphere soils in five regions of Zhejiang Province with severe bacterial blight and performed amplicon sequencing (fungi and bacteria) and metagenomic sequencing. We systematically investigated the impact of phyllosphere diseases on the assembly and function of rhizosphere microbial communities through methods such as amplicon sequencing (fungi and bacteria) and metagenomic sequencing. This research report sheds new light on the microbial ecological mechanisms that rhizosphere microbial communities use to respond to BLB.

## MATERIALS AND METHODS

### Plant and sample collection

All samples used in this experiment were collected between 14 August and 16 September 2021. Healthy and diseased rhizosphere samples were collected from five regions namely: Jiangbei District, Ningbo City (121°32′54″E, 29°57′50″N), Yongjia County, Wenzhou City (120°44′14″E, 28°9′38″N), Ruian City (120°44′1″E, 27°50′22″N), Wucheng District, Jinhua City (119°25′10″E, 21°0′51″N), and Huangyan District, Taizhou City (121°17′19″E, 28°34′6″N). A total of 78 rice rhizosphere samples were collected. The diseased rhizosphere samples selected had rice plants showing severe symptoms of bacterial leaf blight, leaf chlorosis, and necrosis (Fig. S1), while the healthy samples showed no symptoms of rice bacterial blight (Fig. S1). In each region, at least six replicate samples were collected and each replicate sample was a mixture of three sub-samples. The rhizosphere samples of each rice plant from the field were collected, and the excess soil was shaken off as much as possible. The root system collected was kept in a 50 mL sterile centrifuge tube and stored in dry ice, which was transported to the laboratory and

saved at −80 ℃ for further experiments. The disease index statistics (in terms of leaves) were conducted according to the grading standard of white leaf blight disease: grade 0: no disease, grade 1: spot area of less than 10% of leaf area, grade 3: spot area of 11 %–25 % of leaf area, grade 5: spot area of 26 %–45 % of leaf area, spot area of 46 %–65 % of leaf area, and spot area of leaf area of 65% or more. The calculation formula used is as follows: incidence rate = spot area/leaf area. Disease index = ∑ (number of diseased leaves at each level × relative level)/(total number of surveyed leaves × highest level value) × 100% (33).

## DNA extraction and amplicon sequencing

Microbial DNA was extracted from rice rhizosphere samples using the E.Z.N.A. Soil DNA Kit (Omega Bio-tek, Norcross, GA, USA) according to the manufacturer's protocols. Bacterial and fungal sequencing libraries from 78 DNA samples were prepared. The V3–V4 region of the bacteria 16S ribosomal RNA gene was amplified by PCR (95℃ for 2 min, followed by 25 cycles at 95℃ for 30 s, 55℃ for 30 s, and 72℃ for 30 s, and a final extension at 72℃ for 5 min) using primers 341F (5′-CCTAYGGGRBGCASCAG-3′) and 806R (5′-GGACTACNNGGGTATCTAAT-3′). The amplification of ITS1-ITS2 region (internal transcribed spacer) was performed using ITS1F (5′- CTTGGTCATTTAGAGGAAGTAA-3′) and ITS2R (5′- GCTGCGTTCTTCATCGATGC-3′) primers. PCR reaction was carried out using the following conditions: 30 s at 98℃; 35 cycles of denaturation at 98℃ for 10 s, annealing at 54℃ for 30 s, and extension at 72℃ for 45 s; and a final step for 10 min at 72℃. Among them, the barcode was an eight-base sequence that was unique to each sample. PCR reactions were performed in triplicate of 20 µL mixture containing 4 µL of 5× FastPfu Buffer, 2 µL of 2.5 mM dNTPs, 0.8 µL of each primer (5 µM), 0.4 µL of FastPfu Polymerase, and 10 ng of template DNA. Amplicons were extracted from 2% agarose gels and purified using the AxyPrep DNA Gel Extraction Kit (Axygen Biosciences, Union City, CA, USA) according to the manufacturer's instructions. Purified PCR products were quantified by Qubit 3.0 (Life Invitrogen) and every 24 amplicons whose barcodes were different were mixed equally. The pooled DNA product was used to construct the Illumina Pair-End library following Illumina's genomic DNA library preparation procedure. Then, the amplicon library was paired-end sequenced (2 × 250) on an Illumina MiSeq platform Shanghai Biozeron Co., Ltd (Shanghai, China) according to the standard protocols.

## Physiochemical properties of the soil

Fresh soil was used for the determination of ammonium ($NH_4^+$-N) and nitrate ($NO_3^-$-N) content. $NH_4^+$-N and $NO_3^-$-N were extracted using 1 M KCl and their concentrations were measured in a continuous flow analytical system (AA3; Seal Analytical GmbH, Norderstedt, Germany). Soil pH was measured in soil:water solution (1:2.5) using a pH meter (Mettler-Toledo, Shanghai, China). Soil-available phosphorus (AP) was extracted with 0.5 M sodium bicarbonate ($NaHCO_3$) at pH 8.5 and analyzed using the molybdenum antimony anticolorimetric method and an ultraviolet-visible spectrophotometer (UV-2600; Shimadzu, Kyoto, Japan). Available nitrogen (AN) was determined using the alkaline permanganate method (34). Briefly, $HNO_3$ was used as the extract solution, and the soil sample was mixed in a ratio of 20:1, which was shaken for 0.5 hour and filtered immediately. The potassium in the solution was directly measured using a flame photometer, and soil total carbon and total nitrogen were determined using a Vario Max elemental analyzer (Vario Max, Elementar, Langenselbold, Hesse, Germany). Soil organic matter content was determined using the potassium dichromate volumetric method.

## Sequence processing

All analyses of sequencing data were performed in QIIME2 software (Quantitative Insights Into Microbial Ecology 2) (35). First, primers were cut using the cutadapt plugin (36). The DADA2 plugins were quality filtered, denoised, and chimeras were removed (37). Besides, taxonomy assignment was done to ASVs (amplicon sequence variants) using the classify learn naive Bayes classifier (38) based on the SILVA 132 (for bacteria) database and UNITE 8.0 (for fungi) database.

## Shotgun metagenomic sequencing and analysis

The metagenomic shotgun sequencing of rice rhizosphere samples was constructed and sequenced at Shanghai Biozeron Co., Ltd (Shanghai, China). All samples were sequenced in paired-end 150 bp (PE150) mode on an Illumina HiSeqX instrument, and raw sequence reads were quality trimmed using Trimmomatic to remove adapter contaminants and low-quality reads (39). Clean data were obtained by comparing the obtained clean sequences with the host plant genome by the BWA mem algorithm to remove host genome contamination and low-quality data. Clean reads were compared with all bacterial, archaeal, fungal, viral, protozoan, and algal genome sequences in the NBCI database using Kraken2 software (v2.1.1) (40). A clean set of sequence reads per sample was generated using MEGAHIT (41), followed by Prodigal (v2.6.3) (42) to predict the open reading frames (ORFs) of the assembled contigs, and all ORFs generated a set of non-redundant genes after clustering using CD-HIT (43). For gene annotation, BLASTX was used to search the the Kyoto Encyclopedia of Genes and Genomes (KEGG) database for unique gene sets to identify proteins to retrieve their functional annotations. According to the KO results of the samples, the specific functions and pathways of each sample were obtained using the annotated gene mapping pathways of the KEGG pathway database. Predicted genes were converted to amino acid sequences using DIAMOND (44) and compared with the Carbohydrate Active Enzymes (CAZy) database (45) and eggNOG database (46).

## Bioinformatics analysis

The beta nearest taxonomic index (βNTI) was calculated using the null model (999 randomizations) to assess the deterministic and stochastic processes of bacterial community assembly in the rice rhizosphere samples, where |βNTI| > 2 and |βNTI| < 2 represent the deterministic and stochastic processes, respectively. The bacterial community assembly process was further divided into five ecological processes based on βNTI and Bray-Curtis-based Raup-Crick Index (RC-Bray) values, namely βNTI > 2: homogenous selection, βNTI < −2: variable selection, |βNTI| < 2 and |RC-Bray| < 0.95: undominated processes, |βNTI| < 2 and RC-Bray <−0.95: homogenizing dispersal, and |βNTI| < 2 and RC-Bray > 0.95: dispersal limitation (47). To assess the main factors influencing the process of bacterial and fungal taxa assembly in healthy and diseased rhizospheres, Mantel tests based on Spearman's correlation coefficients were performed to compare the Euclidean distance matrix and βNTI values for each variable.

The R package "Hmisc" and "graph" were used to calculate the rhizosphere microbial community co-occurrence network for healthy and diseased groups [relative abundance (RA) > 0.5%, significant correlation $P < 0.05$, Spearman coefficient $N > 0.7$ or <−0.7]. Also, network topology domain analysis was conducted based on average degree, average weighted degree, clustering coefficient, and closeness centrality. Visualization was performed using Gephi 0.9.2 (48). Correlations between rhizosphere microbial communities (bacterial and fungal-based ASVs) and soil physicochemical properties were assessed by Mantel tests. In order to obtain the best discriminative performance of taxa in the healthy and diseased rhizosphere, the Random Forest package was used. A total of 80% of the data were used as the training set, and the Random Forest (importance = TRUE, proximity = TRUE) was used as a function to generate a healthy and disease classification model. Cross-validation was performed by the rfcv () function to select appropriate features. The remaining 20% of the data were used as a test set for models with default parameters.

## Statistical analysis

All sequence data statistical analysis was mainly performed using the R package (v3.6.0). Non-parametric statistical tests were performed (Kruskal-Wallis test or Wilcoxon test) to assess alpha diversity and taxonomical differences at various stages. Based on Bray-curtis distances, the beta diversity of microbial community structure was compared

by non-metric multidimensional scaling analysis (NMDS) and PERMANOVA (permutational multivariate analysis of variance) (49) to assess the significant differences of microbial community structure between groups. Analysis of enriched and depleted ASVs in infected and uninfected phyllosphere was done using the DESeq2 package. At the functional level, differentially abundant KO (KEGG orthologs), CAZy gene families, and Clusters of Orthologous Groups of proteins (COG) were identified using DESeq2. Wilcoxon and Kruskal-Wallis tests were used to examine statistical differences between treatments.

## RESULTS

### BLB affects rhizosphere alpha diversity and beta diversity

From the 78 samples collected, 3,569,207 bacterial 16S rRNAs and 3,426,095 fungal ITS2 of high-quality sequences were obtained. These high-quality sequences were assigned to 10,176 fungal ASVs and 37,088 bacterial ASVs. Overall, the Chao1 index of the infected rice roots for bacteria was significantly reduced (Fig. S2A) with a more significant decrease in diversity observed at Taizhou (TZ) and Yongjia (YJ) (Fig. 1A). Fungi,

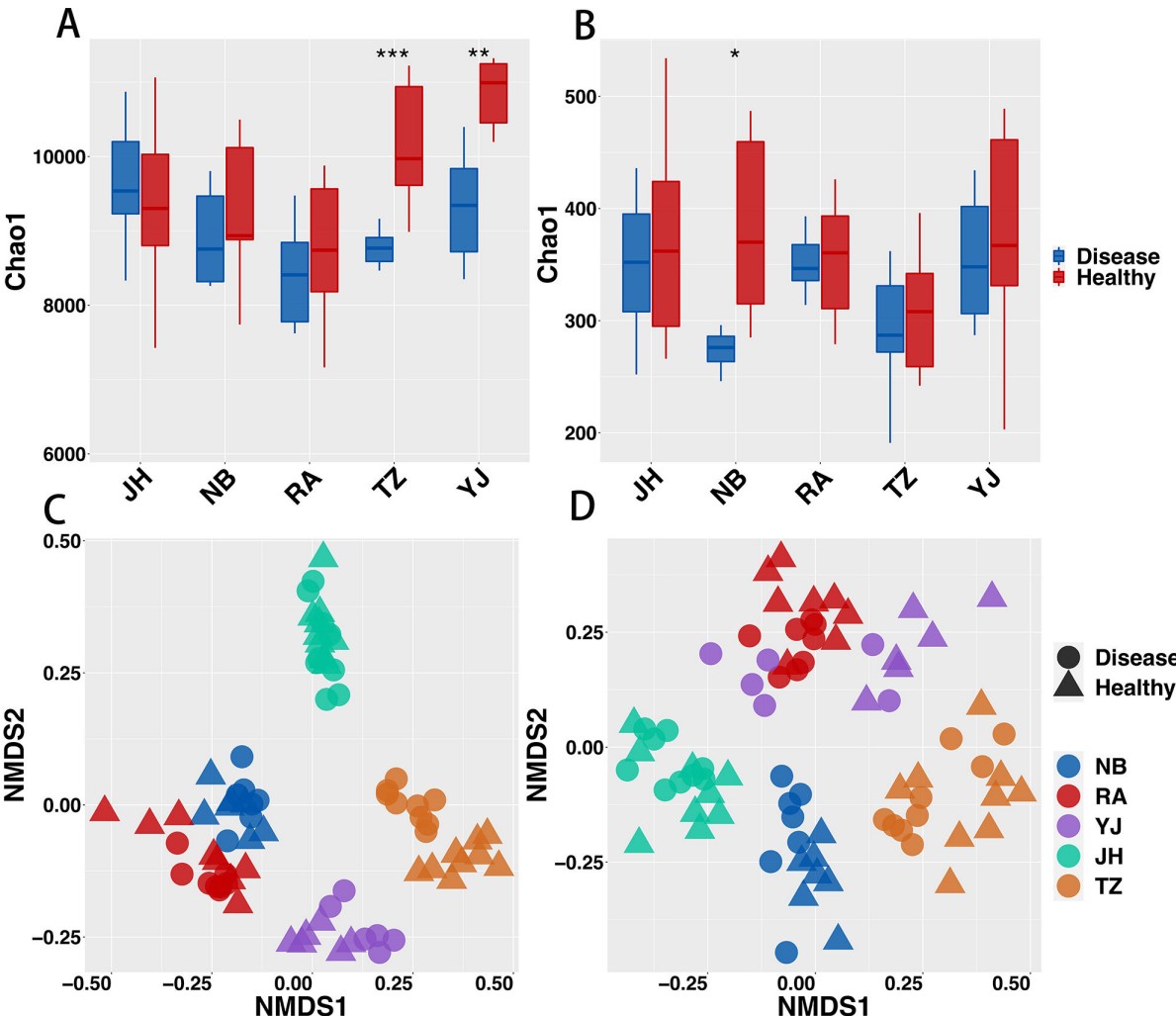

**FIG 1** Alpha diversity and beta diversity of bacterial (A, C) and fungal (B, D) communities in the rhizosphere of healthy and diseased rice. Alpha diversity is characterized based on the Chao1 index. Significant differences were determined using the Kruskal-Wallis test, where asterisks indicate significant differences (*$P$ < 0.05; **$P$ < 0.01; and ***$P$ < 0.001). NMDS is based on Bray-Curtis matrix analysis. Significant differences in communities were tested using PERMANOVA. NB: Ningbo samples, RA: Ruian samples, YJ: Yongjia samples, JH: Jinhua samples, and TZ: Taizhou samples.

on the other hand, did not exhibit any significant difference (Fig. 1B; Fig. S2B). The relative contributions of BLBs and sampling sites to the development of rhizosphere microbial communities were examined. The findings of the NMDS ranking analysis and the PERMANOVA analyses revealed that the sampling site had the greatest significant impact on the change of the rhizosphere microbiome (bacteria: $R2 = 0.54$, $P < 0.05$, fungi: $R2 = 0.38$, $P < 0.05$) (Fig. 1C and D; Fig. S4; Table S1). BLB contributed a higher percentage of change in rhizosphere fungal microbial communities than bacterial communities (bacteria: $R2 = 0.015$, $P < 0.05$; fungi: $R2 = 0.023$, $P < 0.05$) (Fig. 1C and D; Fig. S4; Table S1). Furthermore, the Bray-Curtis dissimilarity and difference index (Fig. S1C and D) was used to evaluate the similarity and the difference between the healthy and diseased rhizosphere microbial communities. It was observed that the microbial communities of the rhizosphere of diseased rice groups were more susceptible to change than the healthy groups. Conclusively, BLB had a substantial impact on the structure and composition of rice rhizosphere microbial communities.

Mantel analysis was used to investigate the relationship between soil physical and chemical properties (Table S5) and to consider whether the changes in the soil physico-chemical properties influenced the structural differentiation of microbial communities. The analysis revealed that the bacterial community in the diseased group was strongly correlated with AK, while the fungal community was strongly correlated with $NO_3^-$-N and AP. Hence, a strong correlation between the bacterial and fungal communities and the AP content in the healthy group was observed (Fig. S3).

## BLB affects rhizosphere microbial community co-occurrence networks

A co-occurrence network analysis was performed to assess the effect of BLB on inter-microbial-fungal-bacterial kingdom interactions in the rice rhizosphere niche. According to our findings, the fungal-bacterial inter-kingdom network pattern changed significantly between two separate ecological niches; healthy and diseased. This revealed that the underlying interactions of the fungi and bacteria differed in the healthy and the diseased rhizosphere microbial networks (Fig. 2A and B). The average degree (13.371), number of edges (2,507), and number of nodes (375) observed in the diseased network were higher than those in the healthy network (average degree: 10.072, number of edges: 1,813, and number of nodes: 360) (Fig. 2C, D, and E). The diseased fungal-bacterial kingdoms exhibited lower network modularity and average path length than the healthy network. However, the inter-kingdom networks of bacteria and fungi showed the reverse pattern (Fig. S5 and S6; Table S2). Additionally, the number of nodes and edges of the bacterial taxa in the diseased network was higher than those in the healthy network, while the reverse pattern was observed in the fungal taxa (Fig. S5 and S6). Bacterial-fungal inter-kingdom networks were 99.5% positively correlated to the healthy group and 96.6% positively correlated to the diseased group. The bacterial intra-kingdom networks were mostly positively correlated (healthy group: 99.6%, diseased group: 96.4%), while the fungal intra-kingdom networks were all positively correlated (Fig. S6; Table S2).

Network keystone taxa are defined as nodes with a high degree (degree > 20) and closeness centrality (closeness centrality > 0.2) in the network. The diseased network comprised 87 network keystone hubs with fungi having 4 network keystone hubs and bacteria having 83 network keystone hubs. These network keystone hubs were primarily composed of *Acidobacteria*, *Chloroflexi*, *Firmicutes*, *Gemmatimonadetes*, *Proteobacteria*, *Mucoromycota*, *Microsporidia*, *Chytridiomycota*, and others (Table S3). Specifically, the diseased-bacteria-fungus inter-kingdom network keystone hubs mainly comprised bacteria like *Streptomyces*, *Sphingomonas*, *Bacillus*, *Gemmatimonas*, and *Cronobacter*, as well as fungi such as *Rhizophydium*, *Mortierella*, and *Diversispora* (Table S3). However, the number of keystone hub species in the diseased rhizosphere co-occurrence network of bacteria (50) was higher than that of the healthy rhizosphere (49), while the number of the keystone hubs in the diseased rhizosphere co-occurrence network of fungi (28) was less than that of the healthy rhizosphere (34) (Table S3).

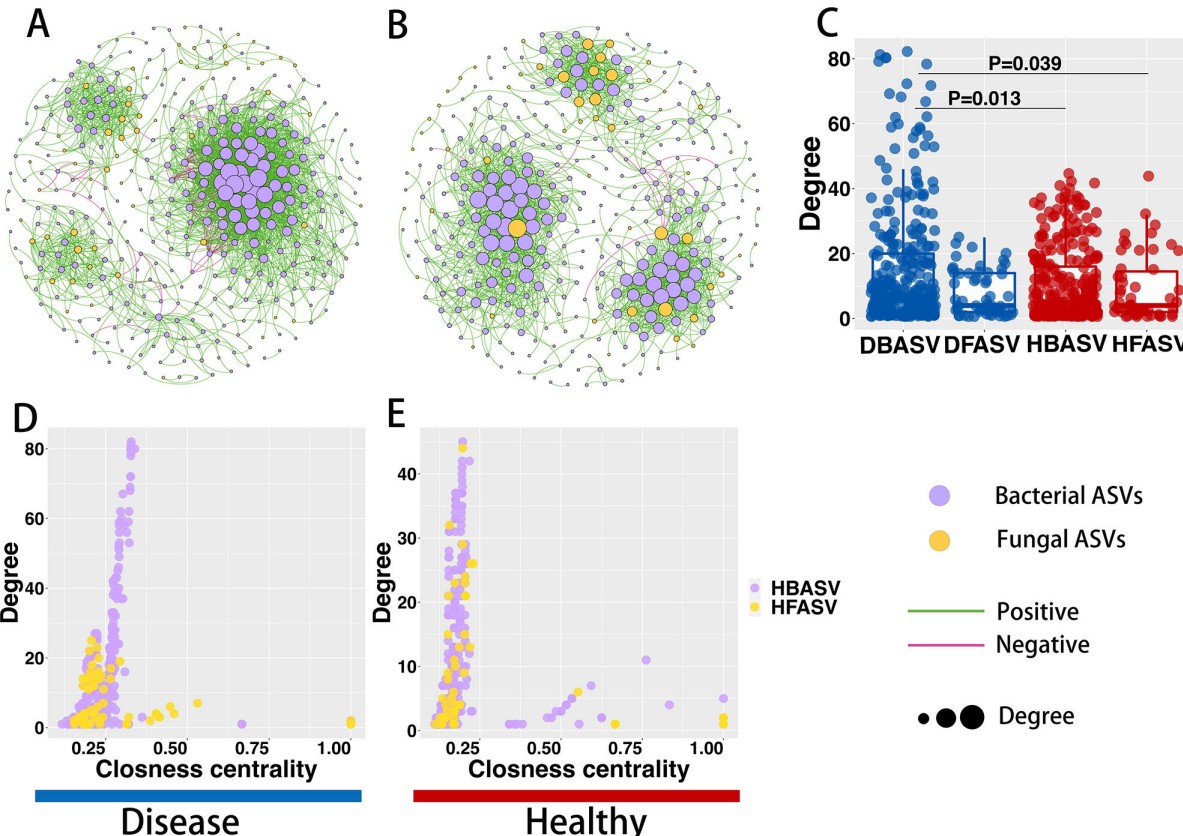

**FIG 2** Co-occurrence networks between fungal and bacterial kingdoms in diseased (A) and healthy (B) rhizospheres. (C) Degree values of bacterial and fungal taxa in healthy and diseased networks. The significant difference was determined by non-parametric Kruskal-Wallis test. Diseased (D) and healthy (E) comparison of networks' node-level topological features (degree and closeness centrality). Node size indicates the network degree. Each node represents an ASV, purple represents bacterial ASV, and yellow represents fungal ASV. Correlations are indicated between nodes (correlation coefficient > 0.7 and the green line indicate a positive correlation; correlation coefficient < −0.7 and the red line indicate a negative correlation).

## BLB affects taxonomic composition in rhizosphere communities

BLB affected the bacterial and fungal composition of the rhizosphere community. The bacteria in all the rhizosphere samples were mostly *Proteobacteria*, *Acidobacteriota*, *Chloroflexi*, *Nitrospirota*, *Desulfobacterota*, *Bacteroidota*, *Actinobacteriota*, *Myxococcota*, Unclassified, and *Gemmatimonadota,* which accounted for 81.7%–86.4% phyla (Fig. 3A). *Ascomycota*, *Mucoromycota*, *Basidiomycota*, *Chytridiomycota*, *Microsporidia*, *Cryptomycota*, *Zoopagomycota*, and *Eukaryota* had no rank and were the most abundant fungi, which accounted for more than 99.9% of all microorganisms (Fig. 3B). The differences in the abundance of bacteria and fungi in the rhizosphere microbes of healthy and diseased rice groups were further compared by DEseq2. The results showed that there were 91 bacterial ASVs (47 ASVs enriched and 44 ASVs depleted) in the rhizosphere, which were significant, while fungi had 30 ASVs that were significantly different (13 ASVs enriched and 17 ASVs depleted) (Fig. S7). Bacteria such as *Rhodanobacter*, *Streptomyces*, *Escherichia*, *Nitrososphaera*, and others, as well as fungi, which mostly included *Nigrospora*, *Slooffia*, *Epicoccum*, and others, were widespread (Fig. S7B). Moreover, further analysis by region revealed that the abundance of *Streptomyces* was significantly increased in the diseased rhizospheres of the four regions (Fig. S7A).

## Rhizosphere bacterial and fungal community assembly

To quantify the deterministic and stochastic processes of microbial community assembly, a combination of βNTI and RC-Bray was used. Variable selection of 70.16% in the

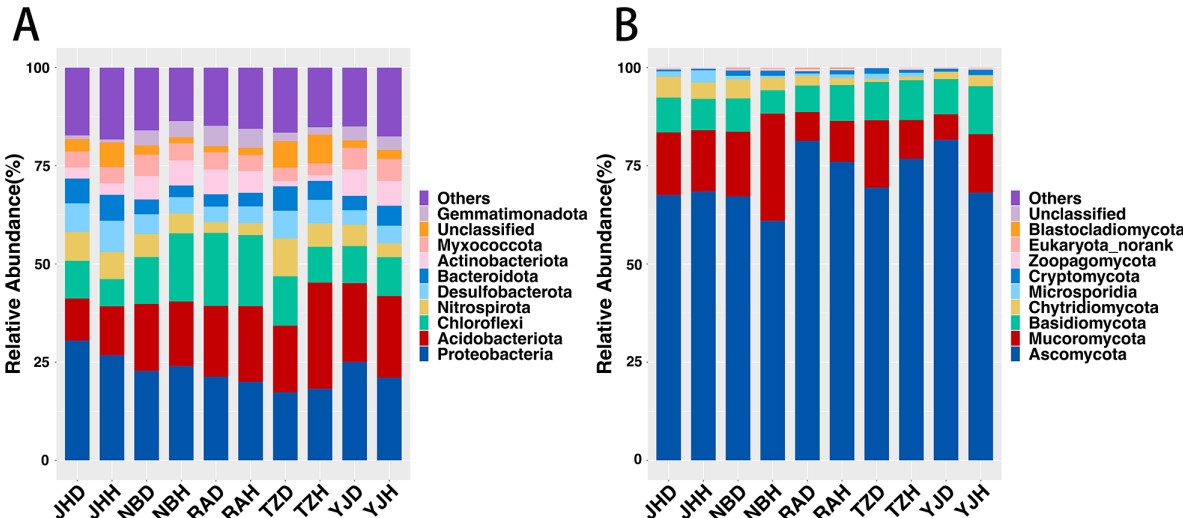

**FIG 3** Taxonomic composition of the rice rhizosphere microbial community. RA of different bacteria (A) and fungi (B) in healthy and diseased. The bacteria or fungi ranked outside the TOP10 were grouped into "Others." NBD: Ningbo diseased samples, NBH: Ningbo healthy samples. RAD: Ruian diseased samples, RAH: Ruian healthy samples. YJD: Yongjia diseased samples, YJH: Yongjia healthy samples. JHD: Jinhua diseased samples, JHH: Jinhua healthy samples. TZD: Taizhou diseased samples, and TZH: Taizhou healthy samples.

deterministic process was primarily responsible for a healthy rhizobacteria microbiome (Fig. 4). In comparison, the fungal community was primarily determined by deterministic variable selection of 47.83%, stochastic undominated processes of 30.89%, and homogeneous selection of 21.28%. (Fig. 4C). The bacterial community in the diseased rhizosphere was primarily determined by deterministic variable selection (87.62%), while the fungal community was determined by variable selection of deterministic processes (44.02%), stochastic undominated processes (13.89%), and homogeneous selection (52.78%). Overall, the bacterial community in the healthy rhizosphere was determined primarily by deterministic process (Fig. 4A), while the fungal community was determined by both deterministic and stochastic processes (Fig. 4B). Nevertheless, the diseased groups showed a similar picture. The increased relative importance of deterministic rhizosphere bacterial communities in the diseased group was significant, whereas the relative importance of the fungal deterministic process decreased and the relative importance of stochastic process increased. Furthermore, the disease rhizosphere bacteria in the five regions were mainly determined by the variable selection (66.67%–100%) of the deterministic process. The fungi in the diseased rhizosphere are primarily determined by the variable selection (33.33%–50.00 %) of the deterministic process, the stochastic undominated processes (13.89%–47.62 %), and homogeneous selection (9.52%–52.78 %) (Fig. S8). Furthermore, correlation analysis showed that the bacterial community assembly in the rhizosphere was significantly positively correlated with changes in AP, AN, and pH (Fig. 4E), while the fungal community was significantly negatively correlated with AP (Fig. 4F), and AN was irrelevant. Thus, it should be noted that the disease index was positively correlated to community assembly (Fig. 4E and F). Therefore, this indicates that disease and soil nutrition both had a significant impact on the assembly of the rhizosphere bacterial and fungal communities.

## Rhizosphere bacterial communities as biomarkers of health and disease

We investigated whether the rhizosphere microbial community membership could be used as a biomarker to distinguish between healthy and diseased rice plants. To do this, we constructed a random forest machine learning model that related healthy and diseased groups with rhizosphere microbial community data at the phylum, class, order, family, genus, and ASV levels. The accuracy of this bacteria model for rhizosphere

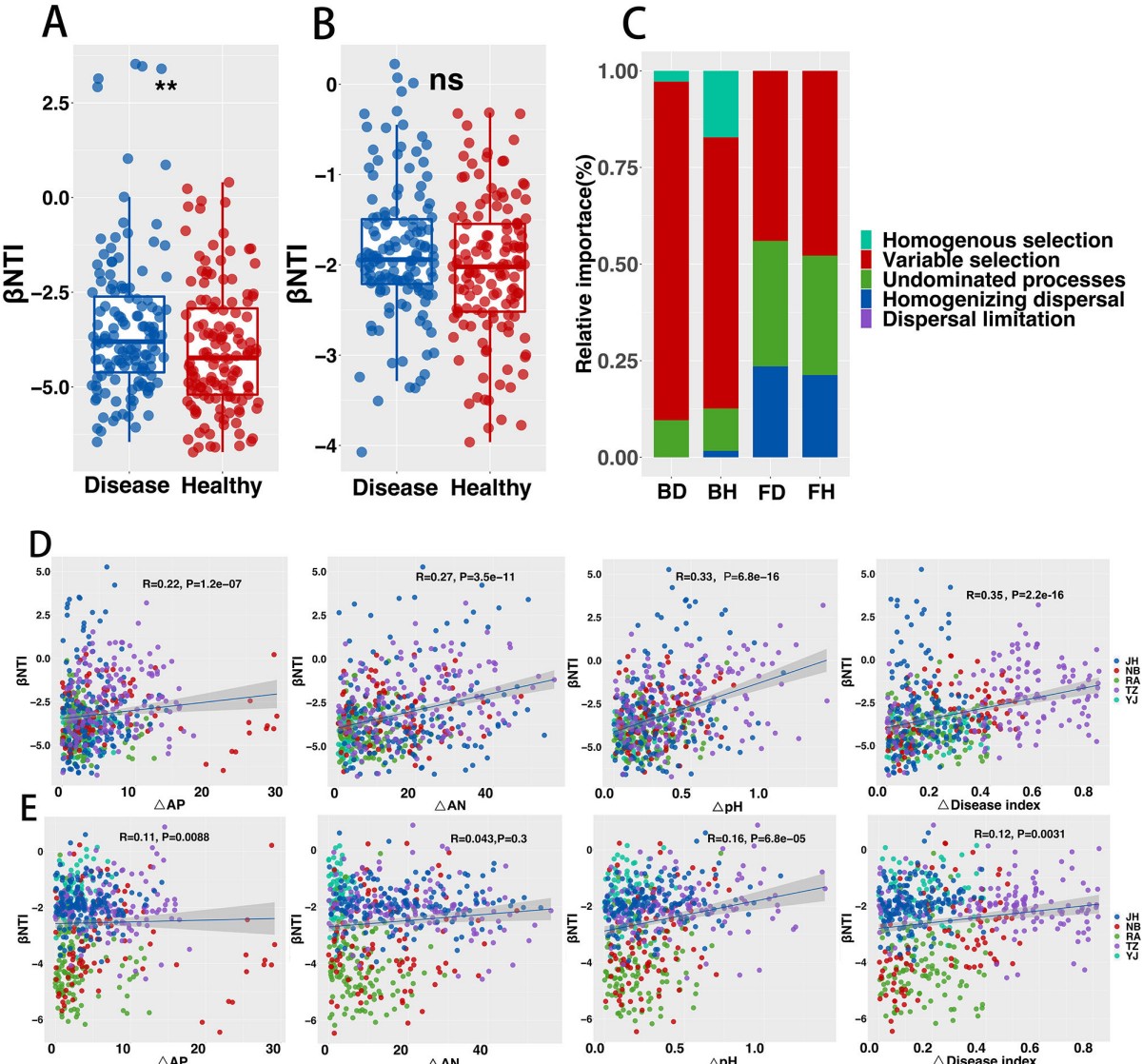

**FIG 4** Assembly processes of healthy and diseased rhizosphere microbial communities (bacteria and fungi). The weighted βNTI of bacterial (A) and fungal (B) communities. (C) Relative importance of bacterial and fungal community assembly processes. Spearman correlations between βNTI and changes in environmental variables for bacterial (D) and fungal (E) communities. The gray dotted lines indicate the upper and lower thresholds for βNTI = 2 and −2, respectively. Asterisks indicate significant differences (**$P$ < 0.01) and ns indicates no significant differences.

microbiota classification was 87.2%, which was the highest among all taxonomic levels. To assess the importance of the indicator microbes, 10-fold cross-validation with five replicates was employed. When 60 significant taxa were used, the cross-validation error curves stabilized (Fig. 5A and C), and when the top significant clusters were chosen, the number of clusters against the cross-validation error curve was very stable. As a result, we categorized these 60 as biomarker clusters. The RA of *Enterobacteriaceae*, *Nostocaceae*, *Micrococcaceae*, and other microorganisms in the diseased rhizosphere was relatively high (Fig. S9). *Acinetobacter* (ASV1770) was the most important biomarker among them (Fig. 5A). However, the average accuracy rate of the test results of the 24 samples that did not participate in the model training in this study was 91.7%, where only two samples were wrong (Fig. 5B). In conclusion, our results indicate that there are consistent differences in the rhizosphere microbiome of healthy and diseased rice plants. These differences can be used to train a random forest machine learning model that can predict whether a rice plant is healthy or diseased with high accuracy. This finding

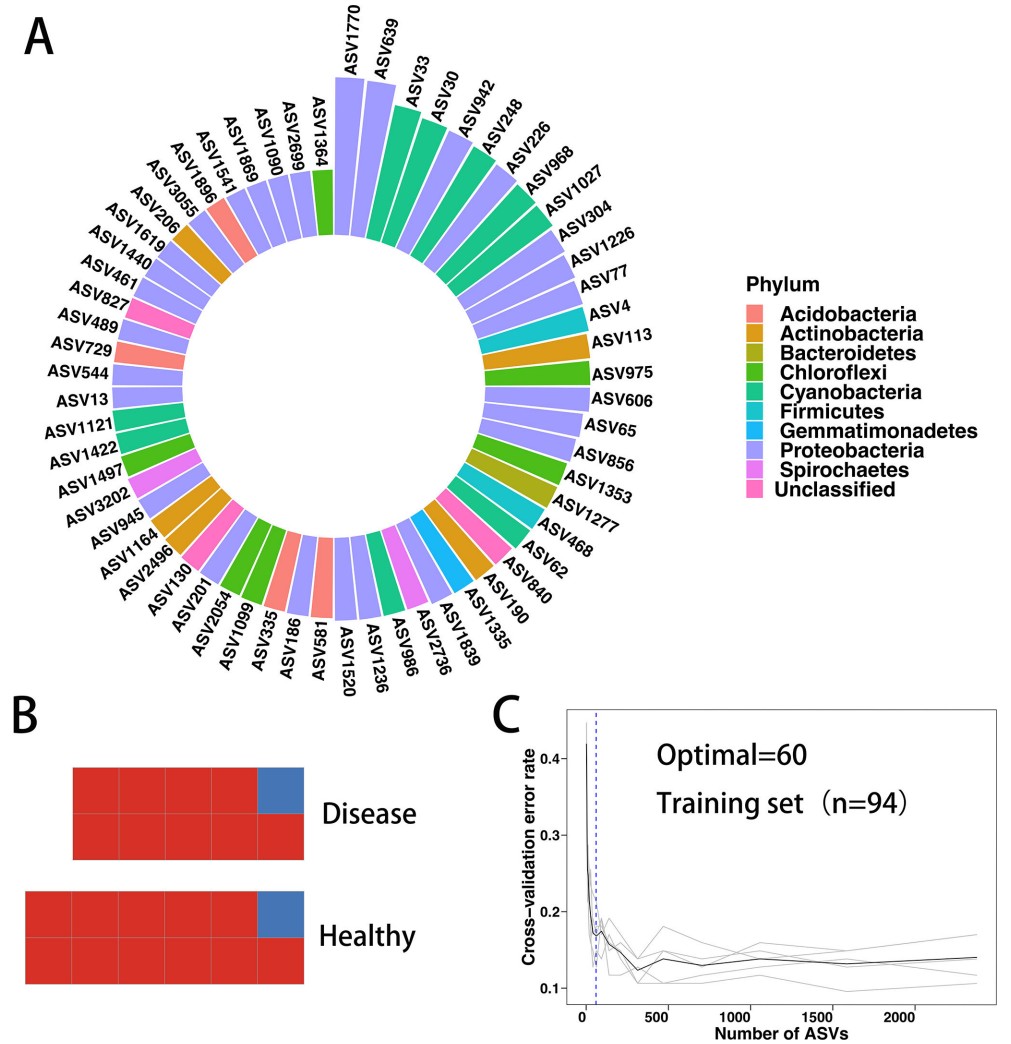

**FIG 5** Random-forest model detects bacterial taxa that accurately predict health and disease. The top 60 biomarkers of importance in the random forest classification model. Using random forest classification, all healthy and diseased rhizosphere microbiomes (94 samples in total) were classified at the ASV level and colored at the phylum level. Biomarker taxa were listed in descending order of importance to model accuracy. (B) Results of the predictions based on the random forest model for healthy and diseased rhizosphere samples were not included in the training set. Each square represents a sample in the test sample. (C) 10-fold cross-validation error as a function of the number of input taxa was used to distinguish between the healthy and diseased root microbiota ordered by importance.

has important implications for the development of new strategies for preventing and controlling BLB.

## BLB affects rice rhizosphere microbiome function

Metagenomic analysis was used to evaluate the functional alteration of the rice rhizosphere microbiome caused by BLB. Samples from the TZ region were selected for metagenomic sequencing. Metagenomes were assigned to 30,229 bacteria and 630 fungi. The functional composition of the rhizosphere microbiome differed substantially between healthy and diseased rhizosphere microbiomes, as evidenced by KEGG Orthology, COG, CAZy, and NMDS ranking of Comprehensive Antibiotic Resistance Database (CARD) (Fig. 6A, B, and C; Table S1). Notably, the BLB rhizosphere microbiome was established on KO. Although the functional diversity of BLB decreased (Fig. 6G),

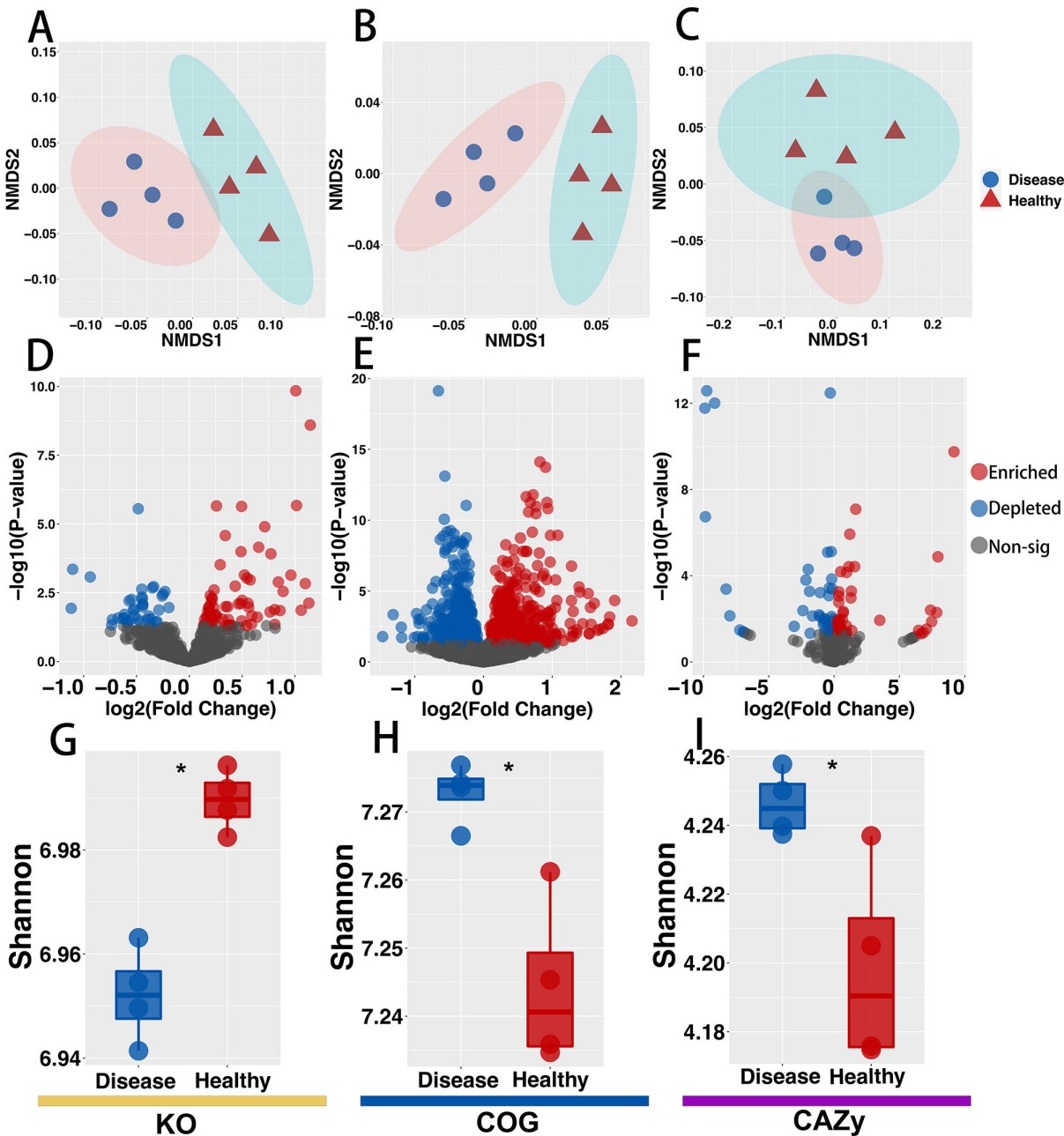

**FIG 6** Functional diversity and differential changes in healthy and diseased rhizosphere microbiomes based on KO, CAZy, and COG functions. KO (A), COG (B), and CAZy (C) NMDS ranking of functional genes based on Bray-Curtis distance matrix. Volcano plots show enrichment and depletion patterns of healthy and diseased rhizosphere microbes based on KO (D), COG (E), and CAZy (F). Boxplots show the functional alpha diversity of the microbiome based on KO (G), COG (H), and CAZy (I) functions. Alpha diversity was based on the Shannon index. Asterisks indicate significant differences (*$P < 0.05$).

functional diversity based on COG and CAZy was higher in the healthy group (Fig. 6H and I).

According to the KEGG database comparison results, 68 KOs were significantly enriched and 39 KOs were depleted in the BLB rhizosphere. The 68 enriched KOs were mostly involved in carbohydrate metabolism (18), cofactor and vitamin metabolism (10), membrane transport (8), energy metabolism (8), amino acid metabolism (6), nucleotide metabolism (4), exogenous biodegradation and metabolism (4), and plant interaction (Fig. 6D). By comparing the rhizosphere microbiome of the diseased group to the CAZy database, it was observed that the diseased group of rhizosphere

microbiome had higher CAZy diversity than the healthy group (Fig. 6I). Differential enrichment findings showed that 41 CAZy families were significantly enriched in the diseased rhizosphere, while 42 CAZy families were significantly depleted (Fig. 6F). The 41 CAZy discovered in the rhizosphere belonged to glycoside hydrolases (21), carbohydrate-binding modules (10), glycosyltransferases (6), and polysaccharide lyases (4). It is worth noting that amylase (beta-amylase) was the diseased group with the highest significant increase (Fig. 6E). This indicates that the diseased rhizosphere microbiome is more capable of behaving as a glycoside hydrolase than healthy ones. Furthermore, when compared to the eggNOG database, 445 COG functional proteins were shown to be significantly increased in the diseased rhizosphere, while 404 COG functional characteristics were decreased. Notably, the most abundant enriched functional protein was Cellular Processes and Signaling (COG_T) (Fig. 6F).

## DISCUSSION

In the present study, the rice rhizosphere microbiome response to BLB was explored using amplicon and metagenomic approaches. The rhizosphere microbiome is well-known to play an important role in plant stress resistance and disease suppression, improving plant tolerance and resistance to environmental stress (51–53). Previous studies have shown that the alpha diversity of rhizosphere microbial communities plays a major role in plant stability and adaptability to stress and invasion. This is because a highly diverse microbial community is more likely to have the necessary traits to resist stress and invasion. Additionally, a diverse community is more likely to be able to adapt to changes in the environment (54–56). However, our findings revealed that the alpha diversity of the diseased rhizosphere bacterial community was lower than that of the healthy rhizosphere (Fig. 1A; Fig. S2A), whereas there was no significant difference in the fungal communities (Fig. 1B; Fig. S2B), which is similar to previous studies (57, 58). Furthermore, our results revealed that the beta diversity of bacterial and fungal communities in the diseased rhizosphere was much lower than that observed in the healthy rhizosphere. In addition, the sampling site had a greater impact on community composition than BLB (Fig. 1C and D; Table S1). Re-analysis of samples from different regions found that BLB can significantly affect the rhizosphere microbial community composition (Fig. S2A, B and S4; Table S1), which indicates that BLB can indeed significantly affect the rhizosphere microbial community structure. Taken together, these findings suggest that BLB significantly decreased the diversity of bacterial communities in the rhizosphere, with a greater impact on the bacterial community than the fungal community.

Co-occurrence network analysis provided novel insights into the complex microbial communities and the interactions within microbial communities. In addition to the differences in bacterial communities' diversity, diseased rhizospheres formed microbial co-occurrence networks with a higher number of edges, nodes, and average degree than healthy plants. However, fungal communities had the opposite pattern (Fig. 2; Fig. S5; Table S2). When microbes are stressed, highly connected co-occurrence networks could form (59) and such complex networks are more favorable to plant growth, hence, enhancing their adaptability to environmental shocks than simple networks. Wei et al. (57) reported that the complexity of the community co-occurrence network in the cotton rhizosphere increased significantly after *Verticillium dahlias* infection (57). In agreement with this study, the bacterial co-occurrence network had a higher proportion of negative edges than the fungal network (Fig. S5; Table S2). Contrary to popular opinion that mutual negative in microbial networks suggests highly intense ecological competition, and the uncertainty caused by such competition will promote network stability and the host thereby benefits from the microbial competition (60). Furthermore, because modularity is strongly associated with network stability, the low modularity of bacterial communities in the diseased rhizosphere exacerbates network instability (61). This was also revealed in this study in that the bacterial community in the rhizosphere was more sensitive to BLB than the fungal community. Our findings suggest that BLB increased

the complexity of bacterial communities while having less influence on fungal networks. Hence, it can be concluded that bacterial communities were more susceptible to BLB.

A substantial difference was observed in the microbial community composition between the BLB rhizosphere and the healthy rhizosphere in terms of species composition (Fig. 3). Potentially beneficial bacteria such as *Streptomyces*, *Chitinophaga*, *Sphingomonas*, and *Bacillus* were more abundant in the BLB rhizosphere (Fig. S7). It has been reported that when plants get infected with pathogens, plants release root secretions (such as amino acids, long-chain organic acids, and so on) in order to attract beneficial microbes that will help the plants combat external environmental stress and boost disease resistance (27, 62–64). Many previous studies have reported that *Streptomyces*, *Chitinophaga*, *Sphingomonas*, and *Bacillus* played an important role in regulating plant physiological states, especially in disease mitigation (15, 65–68). *Streptomyces*, for example, has the capacity to synthesize bioactive chemicals, antibiotics, and extracellular enzymes. It could even contribute to disease resistance by producing antibiotics and siderophores, volatile secretions, and partake in nutrient competition (28, 65, 69). Also, it has been reported that *Streptomyces* can directly promote plant growth by producing phytohormones and enhancing nitrogen fixation (50, 70). *Sphingomonas*, on the other hand, was significantly enriched in the citrus phyllosphere, which enhanced the inhibitory effect on the pathogen, *Diaporthe citri*, through iron competition (66). Similarly, the genus *Bacillus* is a well-known biocontrol bacteria that protect plants from pathogenic bacteria through several mechanisms (such as antibiotic chemical synthesis, biofilm formation, and nutritional niche competition), which is extensively employed in plant disease control (71). Our findings, therefore, imply that when rice is challenged with BLB, the plants select some beneficial bacteria in the rhizosphere. *Streptomyces*, *Sphingomonas*, *Bacillus*, and other genera were considered as the keystone hub taxa of the co-occurrence network through network core species identification (Fig. 2; Table S3). Keystone hub taxa occupy key network topological positions and can be used to construct beneficial plant microbiomes (72). At the same time, these keystone hub taxa play a vital role in network stability (73). Furthermore, more beneficial bacteria that fall in the crucial groups play a core role in promoting plant growth after disease encounters (74). Our studies highlight the importance of these microbial taxa in the rhizosphere network. At the same time, it provides critical evidence for the plant's "cry for help" strategy, which can better harness its microbiome to improve disease control. In the future, it will be necessary to isolate and culture these beneficial bacteria to determine their disease resistance and its molecular mechanism.

According to metagenomic data, BLB significantly reduced the functional diversity of KOs in the rhizosphere microbiome. A decrease in microbial diversity explains the decrease in functional diversity observed. However, the functional diversity of COG and CAZy increased significantly (Fig. 6G, H, and I). Furthermore, it was observed that the functional genes associated with plant-pathogen interaction (K02358, elongation factor Tu) were significantly enriched in the diseased rhizosphere. Different biases have been reported in shotgun metagenomic sequencing (75, 76). Hence, for further research on the significance of tuf enrichment, the changed gene could be sourced from the plant genome, to avoid bias in rice genome filtering. The functional diversity and enrichment in diseased and healthy rhizosphere microbiomes were shown by the metagenomic data. In terms of the functional characteristics, there were significant differences between the diseased and the healthy rhizospheres (Fig. 6). BLB lowered KO functional diversity significantly. A decrease in the rhizosphere microbial diversity might explain the decrease in functional diversity observed. Several studies have also reported that species diversity was critical for ecological function (77–80). A more diverse microbiome ensures that plants engage more extensively in ecological processes. Microbial communities having a high diversity are more stable and have more functional redundancy (81). Various functional categories related to microbial metabolism were significantly enriched in the diseased rhizosphere when compared to the healthy groups

(Table S4), such as carbohydrate metabolism (18), cofactor and vitamin metabolism (10), membrane transport (8), energy metabolism (8), and so on.

This result, however, suggests that BLB strengthened the metabolic pathway-based interactions. Among them, the mannose PTS system (K02793) (Table S4), which is a bacterial PTS found only in bacteria and is responsible for transporting and catalyzing the phosphorylation of extracellular carbohydrates and provides various carbon sources for bacterial metabolism (82), was significantly enriched in the diseased rhizosphere of this study. Furthermore, some fructose bisphosphate aldolases (FBA, K01624) were found to be significantly enriched in the rhizosphere (Table S4). The FBA acts as an ATP source to power ion transport, helps the cyanobacterial cells to enhance their tolerance to salt stress, and regulates several metabolic pathways, as these metabolites originate from glycolysis (83). For example, CII-FBA (sll0018) was shown to increase growth rate and biomass accumulation in *Synechocystis* PCC 6803 cells (83). Notably, 4-hydroxy2-oxopen-tanoic acid aldolase (mhpE, K01666) was significantly enriched (Table S4). This MhpE protein pair has been reported to be essential for siderophore formation (84). In addition, a significant enrichment of the phosphonate transport system substrate-binding protein gene (phnD, K02044) was observed (Table S4), which enhances phosphorus metabolism, in addition to promoting carbon metabolism. It has been shown that targeting the phosphonate ABC transporter substrate-binding protein PhnD enhanced phosphate absorption and biofilm synthesis (85). Furthermore, livH (K01997), an ABC transporter substrate-binding protein, was significantly enriched (Table S4), and livH from *Rhizobium leguminosarum* bv *viciae* can promote N element fixation to regulate plant growth (86).

Furthermore, it was discovered that the relative abundance of the alkaline phosphatase gene phoD (K01113), which is responsible for organophosphate cycling, was reduced in the diseased rhizosphere (Table S4), thus indicating that BLB affected the host phosphorus absorption (87). Despite that, the health and diseased functional enrichment of the rhizosphere microbiome provided evidence for a plant "cry for help" strategy. However, this evidence should be validated through *in vitro* culture experiments with the aim to isolate enriched strains and study their disease resistance and underlying mechanisms. Furthermore, the proposed molecular mechanism for the increased abundance of plant-specific genes in response to biotic stress needs additional investigation. These findings imply that BLB affected the functional taxonomic characteristics of the rice rhizosphere microbiome. Diseases have previously been shown to have a considerable influence on plant microbiome assembly (25, 58). Our results indicate that disease affected the assembly and function of the plant rhizosphere microbiome.

Rice agriculture generates about 7%–17% of worldwide CH4 emissions, which are produced by anaerobic archaea in the rice rhizosphere (88). According to our findings, BLB rhizosphere tetrahydromethanopterin S-methyltransferase subunit H (mtrH, K00584) was considerably enriched in the rhizosphere (Table S4), and the subunit of tetrahydromethanopterin S-methyltransferase was involved in an essential enzyme for methane formation from CO (89). Furthermore, three formylmethanefuran dehydrogenase subunits (fmdA, K00200, fmdB, K00201, fmdC, K00200) were significantly enriched (Table S4). Formylmethanefuran dehydrogenase engages in the first step of the methane hydrogenotrophic pathway, the reduction of $CO_2$ to formate in the first step of acetic acid generation. Methanogenic formate dehydrogenases oxidize formate to $CO_2$ in supplying $CO_2$ for hydrogenotrophic methanogenesis (90, 91). CdhD (K00194) and CdhE (K00197) were also shown to be significantly enriched in the BLB rhizosphere (Table S4). CdhD-CdhE are essential steps in the carbon fixation from CO to generate $CH_4$. In other words, a corrinoid iron-sulfur complex is generated in order to provide a methyl group for the second step reaction (92). Taken together, our findings suggest that BLB may enhance the metabolic activity of archaeal methanogens in the rhizosphere microbiome. Hence, this work significantly extends our understanding of disease-induced rhizosphere microbiome functional assembly.

We combined βNTI with RC-Bray to determine how healthy and diseased bacterial and fungal assembly affects fundamental processes. Four fundamental processes

(diversification, dispersal, selection, and drift) can be promoted in the assembly of microbial communities (93), which are employed to describe microbial assembly processes in various environmental scenarios (94–96). Our findings indicated that deterministic variable selection drives diseased bacterial community assembly, whereas stochastic non-dominant processes with homogenous dispersal drives fungal community assembly (Fig. 4). This shows a substantial microbial selection pressure among diseased rhizosphere bacterial communities, whereas fungi did not. The increased relative importance of deterministic processes to the assembly of bacterial communities could be attributed to increased environmental selection pressure and biotic interactions. For abiotic variables, it was found that the rhizosphere's physicochemical properties, especially AK, AN, and pH, were significantly positively correlated with the assembled communities of bacterial and fungal communities (Fig. 4; Fig. S8), which indicate that these soil nutrients were likely to pass through in the community. Increased selective pressure on the rhizosphere bacterial survival and fitness is regulated by abundant nutrients. Furthermore, when plants are afflicted with pathogens, they will produce metabolites that regulate the activity of the rhizosphere microbial community, and then select the enrichment of beneficial microorganisms in the rhizosphere to manage the community's assembly. Previous studies have shown that plants drive microbial community assembly through the enrichment of rhizosphere metabolites (tocopheryl acetate, citrulline, galactitol, stearylglycerol, and behenic acid) (97). There is also dynamic root exudate distribution that participated in the assembly of wild oat (*Avena barbata*) microbial communities (52). Finally, the existence of BLB may have played a role in the formation of the rhizosphere bacterial community.

Significant differences were observed in the rhizosphere communities between the healthy and diseased (Fig. 5); also, the influence of sample location on the rhizosphere community was considerably larger than the effect of BLB on the rhizosphere. Nonetheless, we were able to identify common features of bacterial populations in BLB rhizosphere soils using the RF model. The RF model applied in this study has had a widespread application in microbial ecology research (98–100). For example, microbial signatures at operational taxonomic unit (OTU) levels could be used to distinguish diseased soils from healthy soils (100), or OTU levels could be used to separate rice genotypes (99).

In our research, we discovered that RF model-based ASV levels were the most accurate at distinguishing between healthy and diseased rhizosphere microbial community signatures (Fig. 5). Furthermore, the model performed well at other taxonomic levels, such as when it was used at a family-level random forest classification model to distinguish indica and japonica root microbial communities (98). In conclusion, our results revealed that the diseased rhizosphere microbial community had consistent characteristics. This suggests that the rhizosphere microbial community can be used as a biomarker to diagnose disease in plants. Further in-depth research on marker bacteria can help to improve the accuracy of this diagnosis. This information can then be used to develop strategies to manage plant diseases.

## Conclusions

The alpha diversity of the BLB rhizobacterial community was reduced in this study, while the fungal community was not reduced. The bacterial community in the rhizosphere was more sensitive to BLB than the fungal community. The relative abundance of potentially beneficial microbes in the rhizosphere of BLB was higher. At the same time, the scale and complexity of the BLB rhizosphere bacterial community co-occurrence network increased, and the core species of the co-occurrence network comprised mainly beneficial bacteria. Furthermore, variations in BLB and soil nutrients especially pH and AP were connected to rhizosphere bacterial community and fungal community assembly. Taken together, these findings not only advance our understanding of microbial responses to environmental changes but also underscore the importance of

harnessing the plant microbiome to promote plant health and support the development of sustainable agricultural ecosystems.

## ACKNOWLEDGMENTS

The work was partially supported by National Natural Science Foundation of China (32072472, 31872017), National Key Research and Development Program of Ningbo (2022Z175), Key Research and Development Program of Zhejiang Province, China (2020C02006, 2019C02006), Shanghai Agriculture Applied Technology Development Program (2021-02-08-00-12-F00771), Agricultural and Social Development Project of Jiangbei District, Ningbo in 2021 (2021B01), Zhejiang Provincial Natural Science Foundation of China (LZ19C140002), Zhejiang Basic Public Welfare Research Plan Project (LGN19C140001), and State Key Laboratory for Managing Biotic and Chemical Threats to the Quality and Safety of Agro-products (2010DS700124-ZZ2014; -KF202101; and -KF202205).

## AUTHOR AFFILIATIONS

[1]State Key Laboratory of Rice Biology and Breeding, Institute of Biotechnology, Zhejiang University, Hangzhou , China
[2]Department of Plant Quarantine, Shanghai Extension and Service Center of Agriculture Technology, Shanghai, China
[3]Taizhou Academy of Agricultural Sciences, Taizhou, China
[4]Ningbo Jiangbei District Agricultural Technology Extension Service Station, Ningbo , China
[5]Institute of Biotechnology, Ningbo Academy of Agricultural Sciences, Ningbo, China
[6]State Key Laboratory for Managing Biotic and Chemical Threats to the Quality and Safety of Agro-products, Institute of Plant Virology, Ningbo University, Ningbo, China

## AUTHOR ORCIDs

Hubiao Jiang 🔟 http://orcid.org/0000-0002-0046-6930
Jianping Chen 🔟 http://orcid.org/0000-0002-3849-4080
Bin Li 🔟 http://orcid.org/0000-0002-4581-4775

## FUNDING

| Funder | Grant(s) | Author(s) |
|---|---|---|
| National Natural Science Foundation of China | 32072472, 31872017 | Bin Li |
| National Key Research and Development Program of Ningbo | 2022Z175 | Bin Li |
| Key Research and Development Program of Zhejiang Province, China | 2020C02006, 2019C02006 | Bin Li |
| Shanghai Agriculture Applied Technology Development Program | 2021-02-08-00-12-F00771 | Bin Li |
| Agricultural and Social Development Project of Jiangbei District, Ningbo in 2021 | 2021B01 | Bin Li |
| Zhejiang Provincial Natural Science Foundation of China | LZ19C140002 | Bin Li |
| Zhejiang Basic Public Welfare Research Plan Project | LGN19C140001 | Bin Li |
| State Key Laboratory for Managing Biotic and Chemical Threats to the Quality and Safety of Agro-products | 2010DS700124-ZZ2014; -KF202101; and -KF202205 | Bin Li |

## AUTHOR CONTRIBUTIONS

Hubiao Jiang, Conceptualization, Data curation, Formal analysis, Investigation, Methodology, Visualization, Writing – original draft, Writing – review and editing | Jinyan Luo, Funding acquisition, Project administration, Supervision | Quanhong Liu, Methodology | Solabomi Olaitan Ogunyemi, Methodology, Writing – review and editing | Temoor Ahmed, Methodology, Writing – review and editing | Bing Li, Methodology | Shanhong Yu, Methodology, Resources | Xiao Wang, Methodology | Chenqi Yan, Methodology | Jianping Chen, Funding acquisition, Methodology, Supervision | Bin Li, Formal analysis, Funding acquisition, Methodology, Supervision, Writing – review and editing

## DATA AVAILABILITY

The data sets that support the findings of this study are available in the SRA at the NCBI with the identifier PRJNA806468.

## ADDITIONAL FILES

The following material is available online.

### Supplemental Material

**Supplemental material (Spectrum01059-23-s0001.docx).** Fig. S1 to S9; Tables S1 to S5.

### Open Peer Review

**PEER REVIEW HISTORY (review-history.pdf).** An accounting of the reviewer comments and feedback.

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
