## [Reviewer comments · Microbiology Spectrum]

Microbiology Spectrum

Rice bacterial leaf blight drive rhizosphere microbial assembly and function adaptation.

Hubiao Jiang, Jinyan Luo, Quanhong Liu, Solabomi Olaitan Ogunyemi, Temoor Ahmed, Bing Li, Shanhong Yu, Xiao Wang, Chenqi Yan, Jianping Chen, and Bin Li

Corresponding Author(s): Hubiao Jiang, Zhejiang University

Review Timeline:

Submission Date:	March 9, 2023
Editorial Decision:	July 22, 2023
Revision Received:	August 21, 2023
Accepted:	August 27, 2023

Editor: Victor Gonzalez

Reviewer(s): The reviewers have opted to remain anonymous.

Transaction Report:

DOI: <https://doi.org/10.1128/spectrum.01059-23>

July 22, 2023

Dr. Hubiao Jiang
Zhejiang University
No. 866, Yuhangtang Road
Hangzhou, Zhejiang
China

Re: Spectrum01059-23 (Rice bacterial leaf blight drive rhizosphere microbial assembly and function adaptation.)

Dear Dr. Hubiao Jiang:

The manuscript has been thoroughly reviewed, and I am sending you the comments provided by the reviewers. Both reviewers agree that the topic addressed is of great interest to the community and significantly contributes to understanding the systemic effects of BLB disease on the rhizosphere community. However, there are several concerns raised by the reviewers that warrant further explanation and, if necessary, modifications in the manuscript. I suggest addressing all the points raised by the reviewers to ensure the quality and comprehensiveness of the study.

Link Not Available

Sincerely,

Victor Gonzalez

Journals Department
Reviewer comments:

Reviewer #1 (Comments for the Author):

Here, the authors present a detailed analysis and comparison of microbiome diversity between Xoo-infected and healthy rhizosphere samples. The analysis presented is thorough and generally well done. The authors are able to draw conclusions about the impact of BLB on the diversity and functionality of both the bacterial and fungal microbiomes. Overall, the study adds to the growing body of literature about the impact of rhizobiome composition and function on plant health and disease

resistance. However, there are a few concerns that should be addressed:

1. Due to the sample collection scheme, in which samples were taken from diseased and healthy fields, it is difficult to determine if the differences seen are due to BLB and to the plants modulating the surrounding rhizosphere communities, or if a pre-existing differences between the communities have allowed BLB infection to take hold. Although the authors do present a few examples from the literature in which pre/post infection studies show that pathogen infection does change microbiome composition and diversity, it is not clear that this is what is happening here. Therefore, I felt that the conclusions were sometimes overstated. For example, line 40 of the abstract ("BLB increased the relative abundance...", Introduction line 113 ("investigate the effect of BLB on the abundance..."), and many other places throughout the manuscript. I would encourage the authors to be mindful of this distinction. I don't think it weakens that paper or the analysis presented to be clear about this.

2. Fig 1 shows that the overall changes in alpha diversity between diseased and healthy samples are primarily driven by sample site and not BLB. Although there are similar trends (decrease from healthy to diseased) across most sites (JH shows a non-significant increase from healthy to diseased), the overall trend is clearly driven by the TZ and YJ sites (bacterial) and the NB site (fungal). Fig S3 clearly shows a correlation between soil/environmental factors and bacterial communities. This leads to two questions. First, do these sites differ significantly from the others (no site-by-site data or ranges for soil/environment data is presented)? Do the conclusions presented hold up if these sites are removed from the dataset? Most of the data is presented in aggregate, and I worry about the broad applicability of the conclusions or if they are specific to these three collection sites. Data showing that these sites are similar to the others, or more recognition of this difficulty would strengthen the manuscript. Place where this could be specifically called out include the discussion of Fig 3 (do these sites show similar changes in abundance/composition?) and Fig 4.

3. Similarly, the functional metagenomic analysis is conducted on only the samples from the TZ region (line 441), which was one of the significantly different sites from Fig 1. Was this site selected in advance, or was it chosen because of the differences in Figure 1? I think either way is acceptable, but the fact that this site is different than the others (potentially) needs to be addressed.

Other concerns:

1. The manuscript would benefit from editing for standard English usage; however the writing does not obscure the main points of the manuscript.
2. Please check the numbering of Figures referred to throughout. Line 274 refers to Fig 2C, D, which seems incorrect. Line 353 refers to Fig S6A, should this be Figure S7? There may be others that I missed.

Reviewer #2 (Comments for the Author):

Summary of Key Findings

This study sheds light on the intricate relationships between phyllosphere diseases like BLB and the rhizosphere microbiome, emphasizing the role of bacteria and fungi to these diseases. Understanding these relationships could provide crucial for developing strategies to enhance plant health and sustainable agricultural practices. While the effect of soil-borne diseases on the rhizosphere microbiome is well-researched, the impact of BLB on the rhizosphere microbiome is not fully explored. These findings highlight that rhizosphere bacteria are more sensitive to BLB than fungi, leading to decreased bacterial community diversity and increased complexity and size of microbial co-occurrence networks. This research provides insight into plant-microbiome interactions and the ecological mechanisms by which rhizosphere microbes respond to phyllosphere diseases. Additional analyses are suggested, as described below.

Major Concerns:

Since this research focuses on a phyllosphere disease and its impact on the rhizosphere, I would suggest expanding on previous studies that relate to it in the introduction. Two such studies are referenced, but without explanation. Expanding on the previous studies will put this current research in better context.

I would suggest additional analyses as follows:

Since rhizosphere microbial communities cluster significantly based on site, a correlation analysis between the rhizosphere communities and nearby soil communities would lend evidence to this conclusion. It would also speak to the influence of nearby soil communities vs. rhizosphere microbes on metadata collected.

Include a correlation analysis between the disease index and physicochemical properties to determine if there is a relationship.

Minor Concerns:

Line 128: Clarify what is meant by sub-samples

In the M&M section titled "Statistical Analysis", differentiate those analyses that were done for the amplicon and MG data. As written, it is not clear.

Line 197- Correct publication no. 90

Line 209- States that the statistical analysis was MAINLY performed using R. If any were not done using R this should be indicated (if appropriate).

Include versions of all software used.

In Figure 3, relative abundance scale needs to be corrected (0- 100%)

Starting on line 181- The SILVA database was used for bacterial taxonomy. Was the UNITE database used for fungal taxonomy?

Staff Comments:

Preparing Revision Guidelines

Please return the manuscript within 60 days; if you cannot complete the modification within this time period, please contact me. If you do not wish to modify the manuscript and prefer to submit it to another journal, please notify me of your decision immediately so that the manuscript may be formally withdrawn from consideration by Microbiology Spectrum.

Response to Editor Comments

Editor comments:

The manuscript has been thoroughly reviewed, and I am sending you the comments provided by the reviewers. Both reviewers agree that the topic addressed is of great interest to the community and significantly contributes to understanding the systemic effects of BLB disease on the rhizosphere community. However, there are several concerns raised by the reviewers that warrant further explanation and, if necessary, modifications in the manuscript. I suggest addressing all the points raised by the reviewers to ensure the quality and comprehensiveness of the study.

Response: We are thankful to the Editor for providing us the opportunity to improve the manuscript. In our revised version, we have addressed all the comments raised by the reviewers and point-by-point response to each comment can be found below. We hope that the revised version of the manuscript would be considered for publication in 'Microbiology Spectrum'.

Response to Reviewer 1 Comments

Here, the authors present a detailed analysis and comparison of microbiome diversity between Xoo-infected and healthy rhizosphere samples. The analysis presented is thorough and generally well done. The authors are able to draw conclusions about the impact of BLB on the diversity and functionality of both the bacterial and fungal microbiomes. Overall, the study adds to the growing body of literature about the impact of rhizobiome composition and function on plant health and disease resistance. However, there are a few concerns that should be addressed:

Response: We are thankful to the reviewer 1 for acknowledging the strength our manuscript and for suggesting the changes for further improvement. The point-by-point responses to each comment from Reviewer 1 are listed below.

Point 1: Due to the sample collection scheme, in which samples were taken from diseased and healthy fields, it is difficult to determine if the differences seen are due to BLB and to the plants modulating the surrounding rhizosphere communities, or if a pre-existing differences between the communities have allowed BLB infection to take hold. Although the authors do present a few examples from the literature in which pre/post infection studies show that pathogen infection does change microbiome composition and diversity, it is not clear that this is what is happening here. Therefore, I felt that the conclusions were sometimes overstated. For example, line 40 of the abstract ("BLB increased the relative abundance...", Introduction line 113 ("investigate the effect of BLB on the abundance..."), and many other places throughout the manuscript. I would encourage the authors to be mindful of this distinction. I don't think it weakens that paper or the analysis presented to be clear about this.

Response: Thanks for the valuable comments. We have now rephrased relevant expression throughout the manuscript to make them clearer and more informative as suggested by the Reviewer 1 (Highlighted in yellow, see line no. 39-42, 105-108, 506-508, 658-659 in manuscript file).

Point 2: Fig 1 shows that the overall changes in alpha diversity between diseased and healthy samples are primarily driven by sample site and not BLB. Although there are

similar trends (decrease from healthy to diseased) across most sites (JH shows a non-significant increase from healthy to diseased), the overall trend is clearly driven by the TZ and YJ sites (bacterial) and the NB site (fungal). Fig S3 clearly shows a correlation between soil/environmental factors and bacterial communities. This leads to two questions. First, do these sites differ significantly from the others (no site-by-site data or ranges for soil/environment data is presented)? Do the conclusions presented hold up if these sites are removed from the dataset? Most of the data is presented in aggregate, and I worry about the broad applicability of the conclusions or if they are specific to these three collection sites. Data showing that these sites are similar to the others, or more recognition of this difficulty would strengthen the manuscript.

Place where this could be specifically called out include the discussion of Fig 3 (do these sites show similar changes in abundance/composition?) and Fig 4.

Response: Thanks for your valuable feedback. We believe that the results of our study are still valid, even though they are primarily driven by sample site. Our study was intended to highlight the impact of BLB on the community, and the site data showed that these sites were not significantly different from other sites, see Table S5. The fact that the overall trend of decreased alpha diversity from healthy to diseased is still present ($P=0.08$), even when the three outlier sites are removed, suggests that BLB is still a significant factor. In addition, the results in Fig. S4 and Table S1 show that BLB can significantly affect the composition of rhizosphere fungal and bacterial communities. Our previous findings demonstrated that the alpha diversity of the rhizosphere microbial community of BLB was lower.

*Fig. 3 shows similar species composition at the phylum level, and we performed differential analyzes of healthy and diseased samples at each site. We concentrated more on compositional changes in healthy and diseased rhizospheres at lower taxonomic levels (e.g., genus level), as bacteria such as *Streptomyces*, *Sphingomonas*, and *Bacillus* showed higher abundance in most of all disease groups.*

Additionally, our findings suggest that bacterial community assembly is primarily a deterministic process, whereas fungal community assembly is largely stochastic. The

fact that there are three sites (NB/RA/YJ) with increased deterministic relative importance in the bacterial community, and three sites (NB/RA/TZ) with increased stochastic relative importance in the fungal community, suggests that BLB contributes to community assembly impact trends are likely to be consistent.

Furthermore, we have limited our study to a small number of sites, and we acknowledge that this could limit the generalizability of our results. However, we believe that the results of our study are still valuable, as they provide new insights into the relationship between BLB and the rhizosphere microbiota. Additionally, we plan to conduct further studies to investigate the impact of sample site on the results of our study in next manuscript. We also plan to collect data from a larger number of sites, which will help to improve the generalizability of our results.

Point 3: Similarly, the functional metagenomic analysis is conducted on only the samples from the TZ region (line 441), which was one of the significantly different sites from Fig 1. Was this site selected in advance, or was it chosen because of the differences in Figure 1? I think either way is acceptable, but the fact that this site is different than the others (potentially) needs to be addressed.

***Response:** We are thankful for the reviewer's thoughtful comments on our work. We selected TZ regions for metagenomic functional analyses, based on the sites with the greatest differences in the results of alpha diversity and beta diversity. Through the analysis of soil physicochemical properties in different regions, it is found that the difference between the TZ region and other regions is not consistent. For details, see Table S5.*

Other concerns:

Point 4: The manuscript would benefit from editing for standard English usage; however the writing does not obscure the main points of the manuscript.

***Response:** Thanks for the valuable suggestion. The manuscript has been carefully revised by a native English Scientist and needful English, grammar, structural, and typographical mistakes have been removed as required by the reviewer 1 (Highlighted in green in manuscript file)*

Point 5: Please check the numbering of Figures referred to throughout. Line 274 refers to Fig 2C, D, which seems incorrect. Line 353 refers to Fig S6A, should this be Figure S7? There may be others that I missed.

Response: Many thanks for your comments. We made corrections and they are highlighted in yellow in the manuscript, see line no 262, 341.

Response to Reviewer 2 Comments

Summary of Key Findings

This study sheds light on the intricate relationships between phyllosphere diseases like BLB and the rhizosphere microbiome, emphasizing the role of bacteria and fungi to these diseases. Understanding these relationships could provide crucial for developing strategies to enhance plant health and sustainable agricultural practices. While the effect of soil-borne diseases on the rhizosphere microbiome is well-researched, the impact of BLB on the rhizosphere microbiome is not fully explored. These findings highlight that rhizosphere bacteria are more sensitive to BLB than fungi, leading to decreased bacterial community diversity and increased complexity and size of microbial co-occurrence networks. This research provides insight into plant-microbiome interactions and the ecological mechanisms by which rhizosphere microbes respond to phyllosphere diseases. Additional analyses are suggested, as described below.

Response: We are thankful to the Reviewer 2 for the careful and constructive evaluation of our paper and for the detailed suggested changes for further improvement of the manuscript. The point-by-point response to the comments of Reviewer 2 can be found below.

Major concerns:

Point 1: Since this research focuses on a phyllosphere disease and its impact on the rhizosphere, I would suggest expanding on previous studies that relate to it in the introduction. Two such studies are referenced, but without explanation. Expanding on the previous studies will put this current research in better context.

***Response:** We completely agree with the reviewer comment, which will make this point clear and understandable. We have made changes as suggested by reviewer 2. (Highlighted in yellow, see line no. 76-84 in manuscript file) .*

Point 2: Since rhizosphere microbial communities cluster significantly based on site, a correlation analysis between the rhizosphere communities and nearby soil communities would lend evidence to this conclusion. It would also speak to the influence of nearby soil communities vs. rhizosphere microbes on metadata collected.

Include a correlation analysis between the disease index and physicochemical properties to determine if there is a relationship.

***Response:** Thanks for your valuable suggestion. We agree that the suggested analyses would be insightful, but unfortunately, we lack nearby soil data in the current study, which limits our ability to conduct the proposed analyses. We appreciate the suggestion and plan to explore these aspects in future experiments or subsequent research papers.*

In addition, we emphasized the effect of BLB on the rice rhizosphere microbial community in this study, and the analysis of each site also found that the rhizosphere microbial community clustered significantly according to health and disease (Fig. S2A, B and S4 and Table S1), which indicated that our different sites The rhizosphere community of was significantly affected by BLB. In addition, correlation analysis showed that there was a significant correlation between the disease index and pH and NO_3^- -N, and the remaining physicochemical properties had no significant correlation with the disease index.

Additionally, we have explicitly addressed these data gaps and provided a thorough discussion of the limitations. Furthermore, we have suggested future research directions that could address these constraints and facilitate a more comprehensive analysis of the rhizosphere and its relationship with nearby soil communities and disease indices.

Minor concerns:

Point 3: Line 128: Clarify what is meant by sub-samples

In the M&M section titled "Statistical Analysis", differentiate those analyses that were done for the amplicon and MG data. As written, it is not clear.

Response: Many thanks for your careful evaluation of our paper. the change has been made (Highlighted in yellow, see line 199 and 225 in the revised manuscript file).

Point 4: Line 197- Correct publication no. 90.

Response: Thanks for is comment. We have revised it in the manuscript and highlighted in yellow; see line 189.

Point 5: Line 209- States that the statistical analysis was MAINLY performed using R.

If any were not done using R this should be indicated (if appropriate).

Include versions of all software used.

Response: Many thanks for your careful comment. All statistical analyzes in this study were performed in R.

Point 6: In Figure 3, relative abundance scale needs to be corrected (0- 100%).

Response: Thanks for the valuable suggestion. This has been corrected in the manuscript (Highlighted in yellow; see Fig. 3).

Point 7: Starting on line 181- The SILVA database was used for bacterial taxonomy.

Was the UNITE database used for fungal taxonomy?

Response: Thanks for the proposed correction. This has been corrected in the manuscript (Highlighted in yellow; see line 179).

August 27, 2023

Dr. Hubiao Jiang
Zhejiang University
No. 866, Yuhangtang Road
Hangzhou, Zhejiang
China

Re: Spectrum01059-23R1 (Rice bacterial leaf blight drive rhizosphere microbial assembly and function adaptation.)

Dear Dr. Hubiao Jiang:

Your manuscript has been accepted, and I am forwarding it to the ASM Journals Department for publication. You will be notified when your proofs are ready to be viewed.

Sincerely,

Victor Gonzalez
Editor, Microbiology Spectrum
